# The Lie Derivative for Measuring Learned Equivariance

**Nate Gruver**[*], **Marc Finzi**[*], **Micah Goldblum**, **Andrew Gordon Wilson**
New York University

## Abstract

Equivariance guarantees that a model's predictions capture key symmetries in data. When an image is translated or rotated, an equivariant model's representation of that image will translate or rotate accordingly. The success of convolutional neural networks has historically been tied to translation equivariance directly encoded in their architecture. The rising success of vision transformers, which have no explicit architectural bias towards equivariance, challenges this narrative and suggests that augmentations and training data might also play a significant role in their performance. In order to better understand the role of equivariance in recent vision models, we apply the Lie derivative, a method for measuring equivariance with strong mathematical foundations and minimal hyperparameters. Using the Lie derivative, we study the equivariance properties of hundreds of pretrained models, spanning CNNs, transformers, and Mixer architectures. The scale of our analysis allows us to separate the impact of architecture from other factors like model size or training method. Surprisingly, we find that many violations of equivariance can be linked to spatial aliasing in ubiquitous network layers, such as pointwise non-linearities, and that as models get larger and more accurate they tend to display more equivariance, regardless of architecture. For example, transformers can be more equivariant than convolutional neural networks after training.

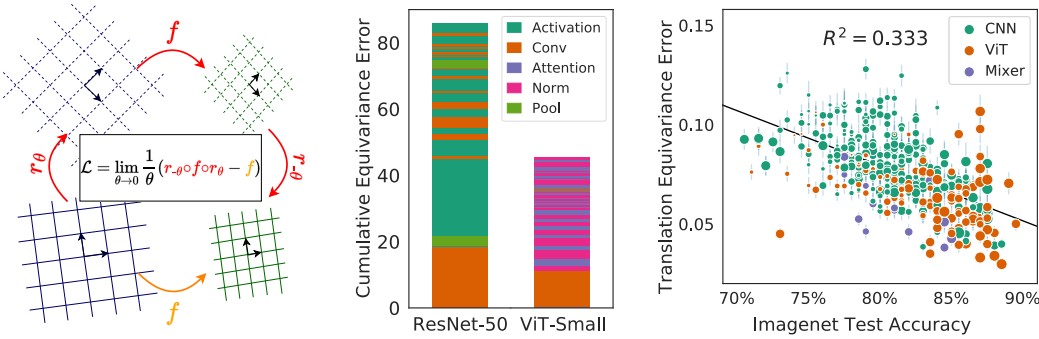

Figure 1: **(Left)**: The Lie derivative measures the equivariance of a function under continuous transformations, here rotation. **(Center)**: Using the Lie derivative, we quantify how much each layer contributes to the equivariance error of a model. Our analysis highlights surprisingly large contributions from non-linearities, which affects both CNNs and ViT architectures. **(Right)**: Translation equivariance as measured by the Lie derivative correlates with generalization in classification models, across convolutional and non-convolutional architectures. Although CNNs are often noted for their intrinsic translation equivariance, ViT and Mixer models are often more translation equivariant than CNN models after training.

## 1 Introduction

Symmetries allow machine learning models to generalize properties of one data point to the properties of an entire class of data points. A model that captures translational symmetry, for example, will have

---

[*]Equal contribution.

the same output for an image and a version of the same image shifted a half pixel to the left or right. If a classification model produces dramatically different predictions as a result of translation by half a pixel or rotation by a few degrees it is likely misaligned with physical reality. Equivariance provides a formal notion of consistency under transformation. A function is equivariant if symmetries in the input space are preserved in the output space.

Baking equivariance into models through architecture design has led to breakthrough performance across many data modalities, including images (Cohen & Welling, 2016; Veeling et al., 2018), proteins (Jumper et al., 2021) and atom force fields (Batzner et al., 2022; Frey et al., 2022). In computer vision, translation equivariance has historically been regarded as a particularly compelling property of convolutional neural networks (CNNs) (LeCun et al., 1995). Imposing equivariance restricts the size of the hypothesis space, reducing the complexity of the learning problem and improving generalization (Goodfellow et al., 2016).

In most neural networks classifiers, however, true equivariance has been challenging to achieve, and many works have shown that model outputs can change dramatically for small changes in the input space (Azulay & Weiss, 2018; Engstrom et al., 2018; Vasconcelos et al., 2021; Ribeiro & Schön, 2021). Several authors have significantly improved the equivariance properties of CNNs with architectural changes inspired by careful signal processing (Zhang, 2019; Karras et al., 2021), but non-architectural mechanisms for encouraging equivariance, such as data augmentations, continue to be necessary for good generalization performance (Wightman et al., 2021).

The increased prominence of non-convolutional architectures, such as vision transformers (ViTs) and mixer models, simultaneously demonstrates that explicitly encoding architectural biases for equivariance is not necessary for good generalization in image classification, as ViT models perform on-par with or better than their convolutional counterparts with sufficient data and well-chosen augmentations (Dosovitskiy et al., 2020; Tolstikhin et al., 2021). Given the success of large flexible architectures and data augmentations, it is unclear what clear practical advantages are provided by explicit architectural constraints over learning equivariances from the data and augmentations. Resolving these questions systemically requires a unified equivariance metric and large-scale evaluation.

In what follows, we introduce the Lie derivative as a tool for measuring the equivariance of neural networks under continuous transformations. The *local equivariance error* (LEE), constructed with the Lie derivative, makes it possible to compare equivariance across models and to analyze the contribution of each layer of a model to its overall equivariance. Using LEE, we conduct a large-scale analysis of hundreds of image classification models. The breadth of this study allows us to uncover a novel connection between equivariance and model generalization, and the surprising result that ViTs are often more equivariant than their convolutional counterparts after training. To explain this result, we use the layer-wise decomposition of LEE to demonstrate how common building block layers shared across ViTs and CNNs, such as pointwise non-linearities, frequently give rise to aliasing and violations of equivariance.

We make our code publicly available at https://github.com/ngruver/lie-deriv.

## 2 BACKGROUND

**Groups and equivariance** Equivariance provides a formal notion of consistency under transformation. A function $f : V_1 \rightarrow V_2$ is equivariant to transformations from a symmetry group $G$ if applying the symmetry to the input of $f$ is the same as applying it to the output

$$\forall g \in G : \quad f(\rho_1(g)x) = \rho_2(g)f(x), \tag{1}$$

where $\rho(g)$ is the *representation* of the group element, which is a linear map $V \rightarrow V$.

The most common example of equivariance in deep learning is the translation equivariance of convolutional layers: if we translate the input image by an integer number of pixels in $x$ and $y$, the output is also translated by the same amount, ignoring the regions close to the boundary of the image. Here $x \in V_1 = V_2$ is an image and the representation $\rho_1 = \rho_2$ expresses translations of the image. The translation *invariance* of certain neural networks is also an expression of the equivariance property, but where the output vector space $V_2$ has the trivial $\rho_2(g) = I$ representation, such that model outputs are unaffected by translations of the inputs. Equivariance is therefore a much richer framework, in which we can reason about representations at the input and the output of a function.

**Continuous signals** The inputs to classification models are discrete images sampled from a *continuous* reality. Although discrete representations are necessary for computers, the goal of classification models should be learning functions that generalize in the real world. It is therefore useful to consider an image as a function $h : \mathbb{R}^2 \to \mathbb{R}^3$ rather than a discrete set of pixel values and broaden the symmetry groups we might consider, such as translations of an image by vector $\boldsymbol{b} \in \mathbb{R}^2$, rather than an integer number of pixels.

Fourier analysis is a powerful tool for understanding the relationship between continuous signals and discrete samples by way of frequency decompositions. Any $M \times M$ image, $h(\boldsymbol{x})$, can be constructed from its frequency components, $H_{nm}$, using a $2d$ Fourier series, $h(\boldsymbol{x}) = \frac{1}{2\pi} \sum_{n,m} H_{nm} e^{2\pi i \boldsymbol{x} \cdot [n,m]}$ where $\boldsymbol{x} \in [0,1]^2$ and $n, m \in [\text{-}M/2, \text{-}M/2 + 1, ..., M/2]$, the bandlimit defined by the image size.

**Aliasing** Aliasing occurs when sampling at a limited frequency $f_s$, for example the size of an image in pixels, causes high frequency content (above the Nyquist frequency $f_s/2$) to be converted into spurious low frequency content. Content with frequency $n$ is observed as a lower frequency contribution at frequency

$$\text{Alias}(n) = \left\{ \begin{array}{ll} n \bmod f_s & \text{if } (n \bmod f_s) < f_s/2 \\ (n \bmod f_s) - f_s & \text{if } (n \bmod f_s) > f_s/2 \end{array} \right\}. \tag{2}$$

If a discretely sampled signal such as an image is assumed to have no frequency content higher than $f_s$, then the continuous signal can be uniquely reconstructed using the Fourier series and have a consistent continuous representation. But if the signal contains higher frequency content which gets aliased down by the sampling, then there is an ambiguity and exact reconstruction is not possible.

**Aliasing and equivariance** Aliasing is critically important to our study because it breaks equivariance to continuous transformations like translation and rotation. When a continuous image is translated its Fourier components pick up a phase:

$$h(\boldsymbol{x}) \mapsto h(\boldsymbol{x} - \boldsymbol{b}) \implies H_{nm} \mapsto H_{nm} e^{-2\pi i \boldsymbol{b} \cdot [n,m]}.$$

However, when an aliased signal is translated, the aliasing operation A introduces a scaling factor:

$$H_{nm} \mapsto H_{nm} e^{-2\pi i (b_0 \text{Alias}(n) + b_1 \text{Alias}(m))}$$

In other words, aliasing causes a translation *by the wrong amount*: the frequency component $H_{nm}$ will effectively be translated by $[(\text{Alias}(n)/n)b_0, (\text{Alias}(m)/m)b_1]$ which may point in a different direction than $\boldsymbol{b}$, and potentially even the opposite direction. Applying shifts to an aliased image will yield the correct shifts for true frequencies less than the Nyquist but incorrect shifts for frequencies higher than the Nyquist. Other continuous transformations, like rotation, create similar asymmetries.

Many common operations in CNNs can lead to aliasing in subtle ways, breaking equivariance in turn. Zhang (2019), for example, demonstrates that downsampling layers causes CNNs to have inconsistent outputs for translated images. The underlying cause of the invariance is aliasing, which occurs when downsampling alters the high frequency content of the network activations. The $M \times M$ activations at a given layer of a convolutional network have spatial Nyquist frequencies $f_s = M/2$. Downsampling halves the size of the activations and corresponding Nyquist frequencies. The result is aliasing of all nonzero content with $n \in [M/4, M/2]$. To prevent this aliasing, Zhang (2019) uses a local low pass filter (Blur-Pool) to directly remove the problematic frequency regions from the spectrum.

While studying generative image models, Karras et al. (2021) unearth a similar phenomenon in the pointwise nonlinearities of CNNs. Imagine an image at a single frequency $h(\boldsymbol{x}) = \sin(2\pi \boldsymbol{x} \cdot [n,m])$. Applying a nonlinear transformation to $h$ creates new high frequencies in the Fourier series, as illustrated in Figure 2. These high frequencies may fall outside of the bandlimit, leading to aliasing. To counteract this effect, Karras et al. (2021) opt for smoother non-linearities and perform upsampling before calculating the activations.

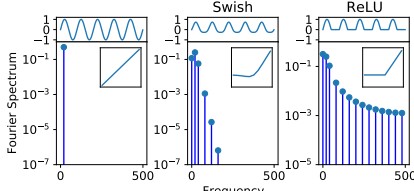

Figure 2: Non-linearities generate new high-frequency harmonics.

## 3 RELATED WORK

While many papers propose architectural changes to improve the equivariance of CNNs (Zhang, 2019; Karras et al., 2021; Weiler & Cesa, 2019), others focus purely on measuring and understanding how equivariance can emerge through learning from the training data (Lenc & Vedaldi, 2015). Olah et al. (2020), for example, studies learned equivariance in CNNs using model inversions techniques. While they uncover several fascinating properties, such as rotation equivariance that emerges naturally without architectural priors, their method is limited by requiring manual inspection of neuron activations. Most relevant to our work, Bouchacourt et al. (2021) measure equivariance in CNNs and ViTs by sampling group transformations. Parallel to our findings, they conclude that data and data augmentation play a larger role than architecture in the ultimate equivariance of a model. Because their study is limited to just a single ResNet and ViT architecture, however, they do not uncover the broader relationship between equivariance and generalization that we show here.

Many papers introduce consistency metrics based on sampling group transformations (Zhang, 2019; Karras et al., 2021; Bouchacourt et al., 2021), but most come with significant drawbacks. When translations are sampled with an integer number of pixels (Zhang, 2019; Bouchacourt et al., 2021), aliasing effects will be completely overlooked. As a remedy, (Karras et al., 2021) propose a subpixel translation equivariance metric (EQ-T$_{\text{frac}}$) that appropriately captures aliasing effects. While this metric is a major improvement, it requires many design decisions not required by LEE, which has relatively few hyperparameters and seamlessly breaks down equivariance across layers. Relative to these other approaches, LEE offers a unifying perspective, with significant theoretical and practical benefits.

## 4 MEASURING LOCAL EQUIVARIANCE ERROR WITH LIE DERIVATIVES

**Lie Derivatives**   The Lie derivative gives a general way of evaluating the degree to which a function $f$ violates a symmetry. To define the Lie derivative, we first consider how a symmetry group element can transform a function by rearranging Equation 1:

$$\rho_{21}(g)[f](x) = \rho_2(g)^{-1} f(\rho_1(g)x).$$

The resulting linear operator, $\rho_{21}(g)[\cdot]$, acts on the vector space of functions, and $\rho_{21}(g)[f] = f$ if the function is equivariant. Every continuous symmetry group (Lie group), $G$, has a corresponding vector space (Lie algebra) $\mathfrak{g} = \text{Span}(\{X_i\}_{i=1}^d)$, with basis elements $X_i$ that can be interpreted as vector fields $\mathbb{R}^n \to \mathbb{R}^n$. For images, these vector fields encode infinitesimal transformations $\mathbb{R}^2 \to \mathbb{R}^2$ over the domain of continuous image signals $f : \mathbb{R}^2 \to \mathbb{R}^k$. One can represent group elements $g \in G$ (which lie in the connected component of the identity) as the *flow* along a particular vector field $\Phi_Y^t$, where $Y = \sum_i a_i X_i$ is a linear combination of basis elements. The flow $\Phi_Y^t(x_0)$ of a point $x_0$ along a vector field $Y$ by value $t$ is defined as the solution to the ODE $\frac{dx}{dt} = Y(x)$ at time $t$ with initial value $x_0$. The flow $\Phi_Y^t$ smoothly parameterizes the group elements by $t$ so that the operator $\rho_{21}(\Phi_Y^t)[\cdot]$ connects changes in the space of group elements to changes in symmetry behavior of a function.

The Lie derivative of the function $f$ is the derivative of the operator $\rho_{21}(g)$ at $g = \text{Identity} = \Phi_0$ along a particular vector field $Y$,

$$\mathcal{L}_Y(f) = \lim_{t \to 0} \frac{1}{t}(\rho_{21}(\Phi_Y^t)[f] - f) = \frac{d}{dt}\bigg|_{t=0} \rho_{21}(\Phi_Y^t)[f]. \tag{3}$$

Intuitively, the Lie derivative measures the sensitivity of a function to infinitesimal symmetry transformations. This local definition of equivariance error is related to the typical global notion of equivariance error. As we derive in Appendix B.1, if $\forall i = 1, ..., d : \mathcal{L}_{X_i}(f) = 0$ (and the exponential map is surjective) then $\forall g \in G : f(\rho_1(g)x) = \rho_2(g)f(x)$ and for all $x$ in the domain, and vice versa. In practice, the Lie derivative is only a proxy for strict global equivariance. We note global equivariance includes radical transformations like translation by 75% of an image, which is not necessarily desirable. In section 6 we show that our local formulation of the Lie derivative can capture the effects of many practically relevant transformations.

The Lie derivative of a function with multiple outputs will also have multiple outputs, so if we want to summarize the equivariance error with a single number, we can compute the norm of the Lie

```python
import torch.nn.functional as F
from torch.autograd.functional import jvp

def rotate(imgs, theta):
    """ Rotate images by angle theta and interpolate"""
    m = [[torch.cos(theta), torch.sin(theta), 0],
         [-torch.sin(theta), torch.cos(theta), 0]]
    m = torch.tensor(m)[None].expand(imgs.shape[0], -1, -1)
    return F.grid_sample(imgs, F.affine_grid(m, imgs.size(), True))

def rotation_lie_deriv(model,imgs):
    """ Lie deriv. of model w.r.t. rotation, can be scalar/image"""
    def rotated_model(theta):
        z = model(rotate(imgs,theta))
        img_like = (len(z.shape) == 4) # more complex for ViT/Mixer
        return rotate(z,-theta) if img_like else z
    return jvp(rotated_model, torch.zeros(1,requires_grad=True))[-1]

def e_lee(model,imgs):
    """ Expected equiv. error (E[|Lf|^2]/d_out) w.r.t. rotation"""
    return rotation_lie_deriv(model, imgs).pow(2).mean()
```

Figure 3: Lie derivatives can be computed using automatic differentiation. We show how a Lie derivative for continuous rotations can be implemented in PyTorch (Paszke et al., 2019). The implementation in our experiments differs slightly, for computational efficiency and to pass second-order gradients through *grid_sample*.

derivative scaled by the size of the output. Taking the average of the Lie derivative over the data distribution, we define Local Equivariance Error (LEE),

$$\text{LEE}(f) = \mathbb{E}_{x \sim \mathcal{D}} \|\mathcal{L}_X f(x)\|^2 / \dim(V_2). \tag{4}$$

We provide a Python implementation of the Lie derivative calculation for rotations in Figure 3 as an example. Mathematically, LEE also has an appealing connection to consistency regularization (Athiwaratkun et al., 2018), which we discuss in Appendix C.1.

**Layerwise Decomposition of Lie Derivative** Unlike alternative equivariance metrics, the Lie derivative decomposes naturally over the layers of a neural network, since it satisfies the chain rule. As we show in Appendix B.2, the Lie derivative of the composition of two functions $h : V_1 \to V_2$ and $f : V_2 \to V_3$ satisfies

$$\mathcal{L}_X(f \circ h)(x) = (\mathcal{L}_X f)(h(x)) + df|_{h(x)}(\mathcal{L}_X h)(x), \tag{5}$$

where $df|_{h(x)}$ is the Jacobian of $f$ at $h(x)$ which we will abbreviate as $df$. Note that this decomposition captures the fact that intermediate layers of the network may transform in their own way:

$$f(h(x)) \mapsto \rho_{31}(g)[f \circ g](x) = \rho_3(g)^{-1} f(\rho_2(g)\rho_2(g)^{-1} h(\rho_1(g)x)) = \rho_{32}(g)[f] \circ \rho_{21}(g)[h](x)$$

and the Lie derivatives split up accordingly.

Applying this property to an entire model as a composition of layers $\text{NN}(x) = f_{N:1}(x) := f_N(f_{N-1}(...(f_1(x))))$, we can identify the contribution that each layer $f_i$ makes to the equivariance error of the whole. Unrolling Equation 5, we have

$$\mathcal{L}_X(\text{NN}) = \sum_{i=1}^{N} df_{N:i+1} \mathcal{L}_X f_i. \tag{6}$$

Intuitively, the equivariance error of a sequence of layers is determined by the sum of the equivariance error for each layer multiplied by the degree to which that error is attenuated or magnified by the other layers (as measured by the Jacobian). We evaluate the norm of each of the contributions $df_{N:i+1} \mathcal{L}_X f_i$ to the (vector) equivariance error $\mathcal{L}_X(\text{NN})$ which we compute using autograd and stochastic trace estimation, as we describe in Appendix B.3. Importantly, the sum of norms may differ from the norm of the sum, but this analysis allows us to identify patterns across layers and pinpoint operations that contribute most to equivariance error.

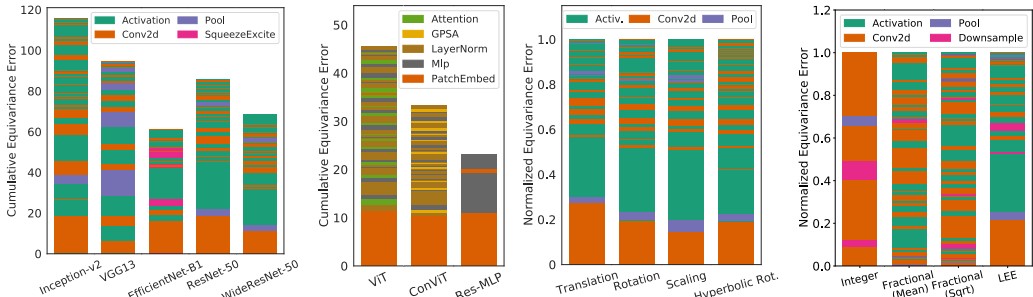

Figure 4: Contributions to equivariance shown cumulatively by layer, in the order the layers occur in the network. **Left**: Convolutional architectures. In all the CNNs, much of the equivariance error comes from downsampling and non-linearities. **Middle-Left**: Non-convolutional architectures. The initial patch embedding, a strided convolution, is the largest contributor for the ViTs and Mixers. The rest of the error is distributed uniformly across other nonlinear operations. **Middle-Right**: ResNet-50 across different transformations as a percentage. Despite being designed for translation equivariance, the fraction of equivariance error produced by each layer is almost identical for other affine transformations, suggesting that aliasing is the primary source of equivariance error. **Right**: Comparing LEE with alternative metrics for translation equivariance. Using integer translations misses key contributors to equivariance errors, such as activations, while using fractional translations can lead to radically different outcomes depending on choice of normalization ($N$ or $\sqrt{N}$). LEE captures aliasing effects and has minimal design decisions.

## 5   LAYERWISE EQUIVARIANCE ERROR

As described in section 3, subtle architectural details often prevent models from being perfectly equivariant. Aliasing can result from careless downsampling or from an activation function with a wide spectrum. In this section, we explore how the Lie derivative uncovers these types of effects automatically, across several popular architectures. We evaluate the equivariance of pretrained models on 100 images from the ImageNet (Deng et al., 2009) test set.

Using the layerwise analysis, we can dissect the sources of translation equivariance error in convolutional and non-convolutional networks as shown in Figure 4 (left) and (middle-left). For the Vision Transformer and Mixer models, we see that the initial conversion from image to patches produces a significant portion of the error, and the remaining error is split uniformly between the other nonlinear layers: LayerNorm, tokenwise MLP, and self-attention. The contribution from these nonlinear layers is seldom recognized and potentially counterintuitive, until we fully grasp the deep connection between equivariance and aliasing. In Figure 4 (middle-right), we show that this breakdown is strikingly similar for other image transformations like rotation, scaling, and hyperbolic rotations, providing evidence that the cause of equivariance error is not specific to translations but is instead a general culprit across a whole host of continuous transformations that can lead to aliasing.

We can make the relationship between aliasing and equivariance error precise by considering the aliasing operation Alias defined in Equation 2.

**Theorem 1.** *For translations along the vector $v = [v_x, v_y]$, the aliasing operation $A$ introduces a translation equivariance error of*

$$\|\mathcal{L}_v(A)(h)\|^2 = (2\pi)^2 \sum_{n,m} H_{nm}^2 \big(v_x^2(\mathrm{Alias}(n) - n)^2 + v_y^2(\mathrm{Alias}(m) - m)^2\big),$$

*where $h(\boldsymbol{x}) = \frac{1}{2\pi} \sum_{n,m} H_{nm} e^{2\pi i \boldsymbol{x} \cdot [n,m]}$ is the Fourier series for the input image $h$.*

We provide the proof in Appendix C.2. The connection between aliasing and LEE is important because aliasing is often challenging to identify despite being ubiquitous (Zhang, 2019; Karras et al., 2021). Aliasing in non-linear layers impacts all vision models and is thus a key factor in any fair comparison of equivariance.

As alternative equivariance metrics exist, it is natural to wonder whether they can also be used for layerwise analysis. In Figure 4 (right), we show how two equivariance metrics from Karras et al. (2021)

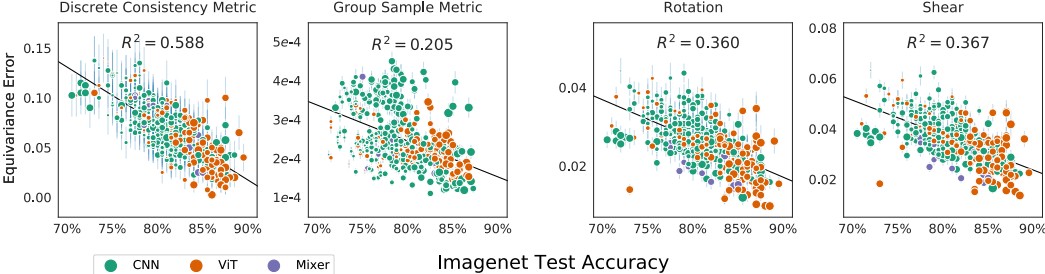

Figure 5: Equivariance metrics evaluated on the ImageNet test set. **Left:** Non-LEE equivariance metrics display similar trends to Figure 1, despite using larger, multi-pixel transformations. **Right:** Norm of rotation and shear Lie derivatives. Across all architectures, models with strong generalization become more equivariant to many common affine transformations. Marker size indicates model size. Error bars show one standard error over test set images used in the equivariance calculation.

compare with LEE, highlighting notable drawbacks. (1) Integer translation equivariance completely ignores aliasing effects, which are captured by both LEE and fractional translations. (2) Though fractional translation metrics (EQ-$T_{frac}$) correctly capture aliasing, comparing the equivariance of layers with different resolutions ($C \times H \times W$) requires an arbitrary choice of normalization. This choice can lead to radically different outcomes in the perceived contribution of each layer and is not required when using LEE, which decomposes across layers as described in section 4. We provide a detailed description of the baselines in Appendix D.1.

## 6 TRENDS IN LEARNED EQUIVARIANCE

**Methodology** We evaluate the Lie derivative of many popular classification models under transformations including $2d$ translation, rotation, and shearing. We define continuous transformations on images using bilinear interpolation with reflection padding. In total, we evaluate 410 classification models, a collection comprising popular CNNs, vision transformers, and MLP-based architectures (Wightman, 2019). Beyond diversity in architectures, there is also substantial diversity in model size, training recipe, and the amount of data used for training or pretraining. This collection of models therefore covers many of the relevant axes of variance one is likely to consider in designing a system for classification. We include an exhaustive list of models in the Appendix D.3.

**Equivariance across architectures** As shown in Figure 1 (right), the translation equivariance error (Lie derivative norm) is strongly correlated with the ultimate test accuracy that the model achieves. Surprisingly, despite convolutional architectures being motivated and designed for their translation equivariance, we find no significant difference in the equivariance achieved by convolutional architectures and the equivariance of their more flexible ViT and Mixer counterparts when conditioning on test accuracy. This trend also extends to rotation and shearing transformations, which are common in data augmentation pipelines (Cubuk et al., 2020) (in Figure 5 (right)). Additional transformation results included in Appendix D.5.

For comparison, we also evaluate the same set of models using two alternative equivariance metrics: prediction consistency under discrete translation (Zhang, 2019) and expected equivariance under group samples (Finzi et al., 2020; Hutchinson et al., 2021), which is similar in spirit to EQ-$T_{frac}$ (Karras et al., 2021) (exact calculations in Appendix D.4). Crucially, these metrics are slightly less *local* than LEE, as they evaluate equivariance under transformations of up to 10 pixels at a time. The fact that we obtain similar trends highlights LEE's relevance beyond subpixel transformations.

**Effects of Training and Scale** In section 3 we described many architectural design choices that have been used to improve the equivariance of vision models, for example Zhang (2019)'s Blur-Pool low-pass filter. We now investigate how equivariance error can be reduced with non-architectural design decisions, such as increasing model size, dataset size, or training method. Surprisingly, we show that equivariance error can often be significantly reduced without any changes in architecture.

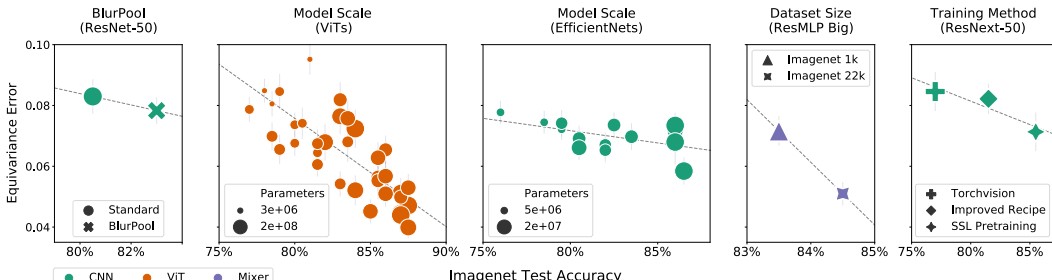

Figure 6: Case studies in decreasing translational equivariance error, numbered left-to-right. **1**: Blur-Pool (Zhang, 2019), an architectural change to improve equivariance, decreases the equivariance error but by less than can be accomplished by improving the training recipe or increasing the scale of model or dataset. **2-3**: Increasing the number of parameters for a fixed model family (here ViTs (El-Nouby et al., 2021) and EfficientNets (Tan & Le, 2019a)). **4**: Increasing the training dataset size for a ResMLP Big (Touvron et al., 2021a) model. **5**: Changing the training recipe for ResNeXt-50 (Xie et al., 2017) with improved augmentations (Wightman et al., 2021) or SSL pretraining (Yalniz et al., 2019). Error bars show one standard error over images in the Lie derivative calculation.

In Figure 6, we show slices of the data from Figure 1 along a shared axis for equivariance error. As a point of comparison, in Figure 6 (left), we show the impact of the Blur-Pool operation discussed above on a ResNet-50 (Zhang, 2019). In the accompanying four plots, we show the effects of increasing model scale (for both ViTs and CNNs), increasing dataset size, and finally different training procedures. Although Zhang (2019)'s architectural adjustment does have a noticeable effect, factors such as dataset size, model scale, and use of modern training methods, have a much greater impact on learned equivariance.

As a prime example, in Figure 6 (right), we show a comparison of three training strategies for ResNeXt-50 – an architecture almost identical to ResNet-50. We use Wightman et al. (2021)'s pretrained model to illustrate the role of an improved training recipe and Mahajan et al. (2018b)'s semi-supervised model as an example of scaling training data. Notably, for a fixed architecture and model size, these changes lead to decreases in equivariance error on par with architectural interventions (BlurPool). This result is surprising when we consider that Wightman et al. (2021)'s improved training recipe benefits significantly from Mixup (Zhang et al., 2017) and CutMix (Yun et al., 2019), which have no obvious connection to equivariance. Similarly, Mahajan et al. (2018b)'s semi-supervised method has no explicit incentive for equivariance.

**Equivariance out of distribution** From our analysis above, large models appear to learn equivariances that rival architecture engineering in the classification setting. When learning equivariances through data augmentation, however, there is no guarantee that the equivariance will generalize to data that is far from the training distribution. Indeed, Engstrom et al. (2019) shows that carefully chosen translations or rotations can be as devastating to model performance as adversarial examples. We find that vision models do indeed have an *equivariance gap*: models are less equivariant on test data than train, and this gap grows for OOD inputs as shown in Figure 7. Notably, however, architectural biases do not have a strong effect on the equivariance gap, as both CNN and ViT models have comparable gaps for OOD inputs.

**Why aren't CNNs more equivariant than ViTs?** Given the deep historical connection between CNNs and equivariance, the results in Figure 5 and Figure 7 might appear counterintuitive. ViTs, CNNs, and Mixer have quite different inductive biases and therefore often learn very different representations of data (Raghu et al., 2021). Despite their differences, however, all of these architectures are fundamentally constructed from similar building blocks–such as convolutions, normalization layers, and non-linear activations which can all contribute to aliasing and equivariance error. Given this shared foundation, vision models with high capacity and effective training recipes are more capable of fitting equivariances already present in augmented training data.

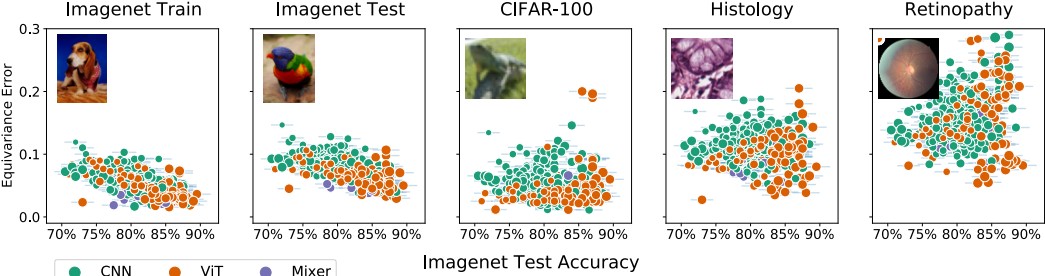

Figure 7: Models are less equivariant on test data and becoming decreasingly equivariant as the data moves away from the training manifold. As examples of data with similar distributions, we show equivariance error on the ImageNet train and test sets as well as CIFAR-100. As examples of out-of-distribution data, we use two medical datasets (which often use Imagenet pretraining), one for Histology (Kather et al., 2016) and one for Retinopathy (Kaggle & EyePacs, 2015).

**Learning rotation equivariance** We finally consider the extent to which large-scale pretraining can match strong architectural priors in a case where equivariance is obviously desirable. We fine-tune a state-of-the-art vision transformer model pretrained with masked autoencoding (He et al., 2021) for 100 epochs on rotated MNIST (Weiler & Cesa, 2019) (details in Appendix D.6). This dataset, which contains MNIST digits rotated uniformly between -180 and 180 degrees, is a common benchmark for papers that design equivariant architectures. In Table 1 we show the test errors for many popular architectures with strict equivariance constrainets alongside the error for our finetuned model. Surprisingly, the finetuned model achieves competitive test accuracy, in this case a strong proxy for rotation invariance. Despite having relatively weak architectural biases, transformers are capable of learning and generalizing on well on symmetric data.

| Model | Test Error (%) |
|---|---|
| G-CNN [13] | 2.28 |
| H-NET [76] | 1.69 |
| ORN [86] | 1.54 |
| TI-Pooling [43] | 1.2 |
| Finetuned MAE | 1.14 |
| RotEqNet [51] | 1.09 |
| E(2)-CNN [73] | 0.68 |

Table 1: Our finetuned MAE is competitive with several architectures explicitly engineered to encode rotation invariance on RotMNIST, where rotation invariance is clearly crucial to generalization.

## 7 CONCLUSION

We introduced a new metric for measuring equivariance which enables a nuanced investigation of how architecture design and training procedures affect representations discovered by neural networks. Using this metric we are able to pinpoint equivariance violation to individual layers, finding that pointwise nonlinearities contribute substantially even in networks that have been designed for equivariance. We argue that aliasing is the primary mechanism for how equivariance to continuous transformations are broken, which we support theoretically and empirically. We use our measure to study equivariance learned from data and augmentations, showing model scale, data scale, or training recipe can have a greater effect on the ability to learn equivariances than architecture.

Many of these results are contrary to the conventional wisdom. For example, transformers can be more equivariant than convolutional neural networks after training, and can learn equivariances needed to match the performance of specially designed architectures on benchmarks like rotated MNIST, despite a lack of explicit architectural constraints. These results suggest we can be more judicious in deciding when explicit interventions for equivariance are required, especially in many real world problems where we only desire approximate and local equivariances.On the other hand, explicit constraints will continue to have immense value when exact equivariances and extrapolation are required — such as rotation invariance for molecules. Moreover, despite the ability to learn equivariances on training data, we find that there is an equivariance *gap* on test and OOD data which persists regardless of the model class. Thus other ways of combating aliasing outside of architectural interventions may be the path forward for improving the equivariance and invariance properties of models.

