# OpenReview forum: "The Lie Derivative for Measuring Learned Equivariance"
_ICLR.cc/2023/Conference — ICLR 2023 notable top 5%_

### Official Review · Reviewer_46iy · 2022-10-22

**Confidence:** 4
**Correctness:** 4
**Technical Novelty And Significance:** 4
**Empirical Novelty And Significance:** 4
**Recommendation:** 8

**Clarity, Quality, Novelty And Reproducibility:**

**Clarity:**
The paper is well-written and it is a pleasure to read. All parts are well-motivated and well-paced, except for the aliasing part in the background section, which is still a bit difficult to grasp without extra reference (But I understand that this is a hard job with limited space).

**Quality:**
This paper studies how to measure equivariance and compare a wide range of existing model architectures in terms of equivariance. It is very important for us to better understand the role of equivariance in deep learning and the relationship between equivariance and generalization. The proposed method is very easy to use (can therefore inspire further investigation) and has a solid theoretical foundation. The empirical results are solid and support their claims well. The main mathematical results seem to be all correct to me.

**Novelty:**
Although there are other metrics to measure equivariance, the proposed local equivariance error based on the Lie derivative has notable advantages (e.g. can analyze the contribution of each layer, local but is connected to global equivariance, very few hyperparameters) and has never been explored in previous literature. Furthermore, there are several very interesting empirical observations in the paper which I think are very inspiring for the community. Although similar findings do exist in other papers, the investigation has never been conducted at such a scale with a wide range of models and training procedures.

**Reproducibility:**
The authors provided the code. Although I did not run the code, lots of experimental details are available from the repository. They also provided some important information about experiments in the text including a simple implementation using automatic differentiation and additional information about empirical study in the appendix.


**Details Of Ethics Concerns:**

I have no ethical concerns over this paper.

**Strength And Weaknesses:**

**Strength:**
1. A simple yet useful technical contribution. Interesting empirical findings that can inspire future research.
2. Solid work with theoretical and empirical support.
3. Nice presentation of the proposed method. Clear motivation.

**Weaknesses:**
1. The explanation for aliasing and equivariance could be more readable to people who are not familiar with signal processing. One solution is that the authors can provide more background knowledge in the appendix, and only keep the high-level essential conclusions and intuition about aliasing in the main text.

**Summary Of The Paper:**

This paper proposes Lie derivative, which is a simple method for measuring equivariance. The authors use the proposed method to analyze the equivalence properties of hundreds of pre-trained models including most popular deep learning architectures for classification. They also demonstrate a layerwise breakdown of the equivariance error for each model, which only becomes possible under their proposed metric. From analysis, they separate the impact of architectures from factors such as model size and training method and identify the contributions of different modules in each model. They also show the interesting finding that large models such as ViT and Mixer are actually "more equivariant" than CNNs which use convolutions with build-in exact equivariance.

**Summary Of The Review:**

I recommend acceptance of this paper mainly based on the following:
1. This paper studies an important question that can inspire further investigations. I think it can benefit both the equivariance community and the broader deep-learning community.
2. The findings in this paper are interesting. The proposed method is simple yet effective. The empirical work is solid.
3. The paper is polished and well-presented.

---

> ### Author Response · Authors · 2022-11-16
> **Author Response: 46iy**
>
> Thank you for your supportive and thoughtful review!
>
> Inspired by your comment, we have added some additional content about aliasing to the appendix. We hope this background material will make the paper more accessible to people with less signal processing experience.

---

> > ### Comment · Reviewer_46iy · 2022-11-21
> > **Response to Authors**
> >
> > Thank you for making the modification. It is great and inspiring work. I will keep my original rating.

---

### Official Review · Reviewer_TkYs · 2022-10-23

**Confidence:** 4
**Correctness:** 3
**Technical Novelty And Significance:** 3
**Empirical Novelty And Significance:** 4
**Recommendation:** 8

**Clarity, Quality, Novelty And Reproducibility:**

Very clearly written.    High quality analysis and  rigorous experiments.  Code included for reproducibility.  The work is novel, but I'd like an added to comparison to a very recent work (LieGG).

**Strength And Weaknesses:**

## Strengths
- The paper considers an important question in the field.  Instead of proposing a novel architecture, this work provides a useful comparison of the strengths and weaknesses of existing state-of-the-art methods from the viewpoint of their ability to learn equivariance.  It could provide insights for improving equivariance in both models with built-in and no built-in inductive bias or help to make model selection easier.
-  The experiments are really quite extensive.  I agree with the authors' claim that the scale and scope of their analysis are helpful in making the trends clear against the noise and other factors contributing to model performance and equivariance error.  Given the goal here is to say something broad about CNNs or ViT in general, it is essential a wide variety of architectures in both classes are considered.
- Previously works on equivariant neural networks have not been overly focused on aliasing.  The equivariance is usually analyzed from the point of view of continuous signals and equivariance is proved in the continuous case.  Empirical measurements are used to assure that discretization errors are acceptable.  However, it seems clear that discretization is important and limiting the potential advantages.  For example, many applications of E(2)-CNN use small discrete rotation groups instead of continuous groups.  This work addresses aliasing head-on and provides evidence of the extent to which it contributes to equivariance error and specifically where (non-linearity).
- The definition of LEE and the observation that LEE can be broken down on a layer-by-layer basis seems like an important contribution. Previously, there was not a precise way to quantify the contribution of each layer to EE.  Practically, one could measure the EE over a single layer (as is done here for alternate EE metrics, I believe), but this is not guaranteed to combine in any reasonable way.
- The result shown in Figure 4 and subsequent insights are very interesting. I appreciate that several different comparisons are done: CNNs vs. ViT, different symmetries, different EE metrics.  The finding that non-linearities contribute the most to LEE raises interesting questions.  What about the fact early layers seem to contribute more?  The observation the contributions to EE for each layer are similar for different symmetry groups suggesting aliasing the root cause is interesting.
- Section 6 discusses several ways to improve equivariance learning given that baking equivariance seems to come up short.


## Weaknesses / Questions
- Since the task is invariant (equivariant with output having trivial rep), the representation must go from having non-trivial group action to trivial group action.  In practice, this invariance can be baked in either by mapping to invariants and having later layers be unconstrained or having equivariant layers and then a final invariant layer.  Thus for a classification task, it is not necessary for the intermediate layers to have low EE for the final NN to be invariant.  I feel this could somewhat undermine the assumptions of the layer-by-layer LEE analysis.  Could the authors comment?  I would feel more confident if the experiments were performed for a task where the output representation was equal to the input representation, for example, image segmentation.
- Another potential aspect overlooked in the setup is the potential for a non-trivial action on the channels of the hidden features.  This absolutely key to making equivariant neural networks work. My understanding is that LEE is assuming the channel representation is trivial (though I could be mistaken).  Thus we could be measuring high LEE when in fact it would be low if we knew the way the channels should transform.
- As noted in the paper, CNNs are often not very equivariant (compared to many other equivariant networks) due to discretization, pooling, and stride. I'd like to see results showing the equivariance error for the CNNs vs. ViTs at initialization and throughout training.  If the compared CNNs do not have low EE at initialization, then the argument that they have built-in equivariance is not very strong.  I'd feel more comfortable with a comparison between E(2)-CNN and unconstrained networks for rotational equivariance error.  In this case, I think the E(2)-CNN architecture would have relatively low EE.
- Although it was posted on the arXiv only in October and has now been accepted to NeurIPS (per the schedule), I believe that LieGG (https://arxiv.org/abs/2210.04345) covers some similar territory in terms of using lie algebra action on functions space to analyze the learned symmetry of the model.  That work also includes analysis of "symmetry bias" (analogous to equivariance error) and per-layer analysis (see Fig.5).  I don't consider it to pre-empt this work, both because they are concurrent and because the focus is very different, however, I would like to see a comparison added.
- I take some issue with the claim that it is surprising that non-equivariance models attain lower equivariance error after training.  Equivariant networks certainly have lower equivariance error at initialization that unconstrained networks, but learning equivariance is necessary in order to get good performance, so it follows that if an unconstrained network achieves better performance than an equivariant one, it may well have lower trained equivariance error (EE).  Consider, for example, a regression task.  Assuming the ground truth is perfectly equivariant, then the equivariance error will be bounded by 2*(approximation error) for any sample by the triangle inequality.   Since you consider a classification task, this relationship is a bit more obscured, but still well represented by your results.

## Questions
- I'm unclear on the aliasing operation $A$ versus the map $\mathrm{Alias}(n)$.  Can you clarify?  In particular, on page 3, it seems $A \colon \mathbb{N} \to \mathbb{N}$, but on page 6, it seems $A \colon (\mathbb{R}^2)^{\mathbb{R}^c} \to (\mathbb{R}^2)^{\mathbb{R}^c}$, i.e. images to images.
- Could section 4 avoid referring to flows and be written in terms of the exponential map $\mathfrak{g} \to G$?
- In section 5, you prove that Alias results in translation LEE.  I didn't fully understand why this points to the non-linear layers.

## Minor Points
- The abstract says "introduce the Lie derivative," I would re-word to make clear Lie derivatives are pre-existing work, they are merely being introduced as a method here.
- I'd replace "it's" with "it is" since this is formal writing.
- In two places $\mapsto$ appears when it should be $\to$.  $\mapsto$ is for elements and $\to$ is for sets.  As in $f \colon X \to Y$ mapping $x \mapsto y$.
- Page 4 "the operator $\rho_{21}(g)[\Phi^t_Y]$ "  Should it be  $\rho_{21}([\Phi^t_Y])$ ?
- In the notation for LEE in eqn. 4, you might want to include the dependence on $X$.
- Section 5, first sentence "equivariance" -> "equivariant"

**Summary Of The Paper:**

This work compares the ability of models with built-in equivariance versus those without in how well they ultimately learn equivariance.  The majority of experiments and analysis focuses on the case of translation equivariance in image classification comparing CNNs to ViT and Mixer, although rotation and scale are also considered.
- In order to measure equivariance error, the authors define local equivariance error, LEE, based on infinitesimal action of the symmetry group on the function space defined by each layer.  This definition allows the authors to decompose the final LEE of the model in terms of per-layer LEE.
- The paper puts forth the hypothesis that the interaction of aliasing, the aberration which result from discretizing a continuous signal, and continuous symmetry is the main driver of equivariance error in all models.   This hypothesis is supported by Theorem 1, which directly computes the equivariance error due to aliasing under translations, and by empirical analysis of layer-wise LEE showing that activation layers contribute most towards LEE.
- Experiments show model performance is correlated with degree of learned equivariance and that there is not a significant difference between CNNs and ViTs in terms of equivariance learning (if anything ViTs have lower LEE.)
- Several architechure-agnostic solutions to better symmetry learning are tested including BlurPool, larger models, more data, and SSL pretraining.  All such interventions (to various degrees) do improve equivariance learning.

**Summary Of The Review:**

I very much liked reading this paper.  A study comparing learned symmetry to built-in symmetry is very useful to the field today. Equivariant architectures are increasing popular and at the same time, non-equivariant methods such as transformers are overtaking them in some domains.  The experiments are extensive and the analysis is clear and rigorous.  I think the conclusions will be useful to practitioners who are choosing between different architectures. I hope the tool of layer-wise LEE analysis could help to improve equivariant neural networks.    I have some disagreement with how some parts of the study are set up, things I would like to do differently, and ultimately with the implication that built-in equivariance is often unnecessary.  However, that is really to say I think this paper will inspire much future research both supporting, extending, and arguing with the hypothesis studied here.

---

> ### Author Response · Authors · 2022-11-16
> **Author Response: TkYs**
>
> Thank you for your supportive and constructive review!
>
> You ask about the extent to which the final classification network can be invariant even when the equivariance error is nonzero for intermediate layers–specifically “for a classification task, it is not necessary for the intermediate layers to have low EE for the final NN to be invariant”. While we can evaluate LEE on an individual layer by considering the input and output of the layer in isolation, in the layerwise analysis we are computing the contribution of the equivariance error to the final equivariance error of the network given by the terms in the sum of eq 6. The distinction lies in the additional Jacobian factor $df_{N:i+1}$, which connects any given layer to the layers that follow it in the network. If the equivariance error at a given layer does not contribute to the final equivariance error, then it will lie in the nullspace of this Jacobian, and so in this sense the layerwise analysis tracks with the final invariance of the network. Importantly, it is strictly possible for the contributed equivariance errors of two layers of the network to point in opposite directions and hence cancel out, and our visualization is only on the norm of the contributions (although the directions could also be compared in a more fine grained analysis).
>
> With respect to your question regarding non-trivial representation on the feature maps (such as in E(2) steerable CNNs and other works), we would like to emphasize a few key points. The value of the end to end equivariance metric is completely independent of the internal representations, and instead depends only on the external representations (image to scalar classification outputs) of the inputs and outputs. This fact can be observed from equation 3 and equation 5. However, while the internal representations have no impact on the end to end metrics, they will affect the informativeness of the layerwise breakdown of the error, as the different terms of equation 6 will differ based on the internal representations (though the sum is the same). For the visualizations of the layerwise breakdown in figure 4, these visualizations are all on networks that have scalars or scalar images as internal representations and so these breakdowns are sensible; however, for networks that do have nontrivial internal representations such as E(2)-CNN, when computing the layerwise breakdown one would need to use the appropriate internal representations that are known for the network and that is part of the LEE methodology (though we only include pseudocode examples for more basic representations).
>
> Thanks for the question about LieGG. We note indeed that this concurrent paper was made public _after_ the ICLR submission deadline, and thus it would be against ICLR policy to have it affect our paper’s evaluation or require a comparison. However, we are happy to include a discussion anyways.The LieGG R function is related to the end to end LEE metric and can be seen as a special case in the scenario where (a) the transformation is a matrix (and therefore the flow field Y(x) is a linear function of x), and (b) the output representation of the function is trivial. On the other hand, the way in which equivariance errors are computed for truncated networks is completely different from our layerwise analysis, where in their method they truncate the network, freeze the weights, train a new classification head with the truncated network, and evaluate the end to end equivariance error with this classification head. In general, the paper has quite a different objective to ours, though it’s interesting work and speaks to the benefits of the Lie derivative and considering equivariance error in a local sense.
>
> To answer your specific technical questions,
> - Re: Alias vs A. The operation A takes the frequency representation of an image as input and returns the frequency spectrum with Alias applied to each frequency component.
> - Re: Reframing section 4 in terms of the exponential map g→G? The flows are just an explicit construction of the exponential map in this case, although we don’t refer to them as such. While we believe it is helpful being more concrete here, we will add a note that this is the exponential map.
> - Re: Aliasing, translation LEE, and non-linear layers. Pointwise nonlinearities have a major effect on LEE that is hard to explain without appealing to aliasing. It’s more immediately obvious how convolutional layers can give rise to boundary effects and spatial distortions that might affect equivariance, but the effects of pointwise non-linearities only become clear when we consider aliasing and the frequency domain.
>
> Many thanks again for your thoughtful and supportive review.
>
> Minor points:
> You are correct about the typos and we have fixed these in the updated draft.

---

> > ### Comment · Reviewer_TkYs · 2022-12-05
> > **Thanks for response**
> >
> > I am thankful to the authors for the clarifying response! I maintain my high score.

---

### Official Review · Reviewer_rDjb · 2022-10-31

**Confidence:** 5
**Correctness:** 3
**Technical Novelty And Significance:** 3
**Empirical Novelty And Significance:** 3
**Recommendation:** 8

**Clarity, Quality, Novelty And Reproducibility:**

The provided code and the text of the paper allow one to reproduce the approach.

**Strength And Weaknesses:**

The paper is well-written. I found it easy to read. The storytelling of hte paper is of high quality. The derivations are accurate and the math notation is well-understood.

**Strengths**
1. The authors pay attention to the problem which is less studied in the community - the problem of measuring the equivariant properties of various models.
2. The theory of the paper is solid. And demonstrates a very good understanding of the problem.
3. The experimental results of the paper demonstrate that such a method is useful for the analysis

**No Strengths**
I don't see any significant issues, which may affect my rating drastically. However, I would like to address several questions and suggestions, which may improve the quality of the paper.
1. The related work misses a section on why we are so motivated to measure the equivariance error. There were several papers which demonstrated that the model's accuracy and its equivariance error are highly correlated. In [1] the authors take a network and train it by minimizing the equivariance error which leads to a better accuracy than any other counterparts. In [2] the authors compare different scale-equivariant models, conclude that the equivariance error is not 0, and demonstrate that the lower it is the more accurate the model becomes.
2. [Does not affect my rating. Just a pure interest] Although it was not possible to include this paper beforehand, there is a recent paper which considers a similar problem. In [3] the authors introduce a similar metric for measuring the symmetric properties of the model. What is the main difference between their approach and yours?
3. Let us consider two neural networks, which perform binary classification. The very last layer outputs a positive or a negative value for either class 1 or class 2. We initialize them with the same weights. However, for network 2 we multiply the output by $10^6$. Do I understand it correctly, that for network 2 LEE will be $10^6$ larger than LEE for network 1? If so, can we really compare two neural networks based on Eq 4? Do we need an extra step of output normalization?
4. In supplementary materials, in A3, paragraph 2, you mention that there is an inequality between the $\mathcal{L}$ of the network and the sum of the $\mathcal{L}$  of its layers. Let us consider a network, for which the very last layer just multiplies everything by $0$. Such a network is absolutely invariant, thus $\mathcal{L} = 0$. However, the $\mathcal{L}$ of the layers of the network can be arbitrary large. Thus, Figure 4 does not really tell us anything about the network as a whole. Could you make a comparison where you plot side-by-side the cumulative loss as a sum per-layer losses. And a series of losses calculated for all truncated subnetworks of the original networks $\text{subnet}_k = f_k \circ \dots \circ f_1$


- [1] Khetan N. et al. Implicit Equivariance in Convolutional Networks, preprint 2021
- [2] Sosnovik I., Moskalev A., Smeulders A. Disco: accurate discrete scale convolutions, BMVC 2021
- [3] Moskalev A. et al. LieGG: Studying Learned Lie Group Generators, NeurIPS 2022

**Summary Of The Paper:**

The paper addresses the problem of measuring to what degree the the function is invariant to certain transformations. The authors suggest to use the Lie Derivative to measure such a quantity. Then Local equivariance error (LEE) is introduced. The paper demonstrates how different parts of various models contribute to the equivariant properties of the model in terms of LEE and how different models demonstrate a correlation between the global LEE and the resulting accuracy.

**Summary Of The Review:**

The paper is worth sharing with the community because of its potential significant contribution to the field. Minor changes may improve its quality.

---

> ### Author Response · Authors · 2022-11-16
> **Author Response: rDjb**
>
> Thank you for your detailed comments and highly supportive feedback!
>
> Thanks for your question about output normalization. When calculating the LEE for classifiers, we are considering the map to the class probabilities rather than the logits, and thus the output scale is fixed by the problem at hand. In general this will be the case with a fixed task, and the output scales should be directly comparable between networks. However, a related point that we did observe is that models could have very different Lie derivative values at initialization (in a way that is not reflective of the equivariance error of the models after training), and this is because at initialization the outputs are not at all constrained by the task. In general we recommend only using LEE to evaluate trained models.
>
> In terms of multiplying a network’s outputs by 0, we note there is no inconsistency between the end-to-end LEE calculation and the contributions of each layer under our decomposition, as the effect on the output will be captured by the Jacobian factor in the layerwise decomposition. However, we found your suggestion of examining the LEE of subnetworks very interesting, and inspired by your comments we performed the experiment in question on a ResNet50 and have included a plot and discussion in the appendix. This experiment provides further evidence that our layerwise breakdown is an appealing solution for quantifying the impact of individual layers on the equivariance of a network’s outputs.
>
> We would like to make a few clarifications regarding motivation and related work. First, there is a notable distinction between evaluating invariance on problems that contain exact invariances (e.g. RotMNIST) and problems with approximate invariances (e.g. ImageNet). While it is obvious that equivariance error should be strongly correlated with accuracy on RotMNIST, this correlation is less obvious in the case of ImageNet, and we were not aware of any work that explored this connection in depth–though there has been contemporaneous work supporting our findings [1]. There are also earlier works that have pointed out the connection between invariance and classification accuracy in natural images, e.g. Zhang 2019 [2]. We build on that observation while providing a more precise set of tools and broader collection of observation for natural images. Importantly, while we focus on ImageNet and related datasets in our results, LEE naturally extends to datasets like RotMNIST and MNIST-scale (the datasets used in the papers you mention) and LEE provides a useful counterpoint to existing metrics (see paragraph 2 of Section 3 in the text).
>
> Thanks for the question about LieGG. We note indeed that this concurrent paper was made public _after_ the ICLR submission deadline, and thus it would be against ICLR policy to have it affect our paper’s evaluation or require a discussion. However, we are happy to include a discussion anyways. We believe LieGG is exciting concurrent related work. LieGG is using the Lie derivative to recover group generators, and the symmetry bias reported in their experimental results is a special subcase of LEE. While the mathematics behind LEE and LieGG is similar, the goals of the corresponding papers are quite different. LieGG seeks to recover group generators on problems where they are known a priori. On the other hand, we largely focus on models trained and evaluated on ImageNet (including in our RotMNIST experiments, where we finetune a model with extensive ImageNet pretraining on the smaller dataset). In this way, LieGG is complementary to the investigation here and further evidence that the Lie derivative is a very important tool that can be useful to researchers working on a variety of problems, including those with both strict and approximate invariances.
>
> [1] Deng et al. On the Strong Correlation Between Model Invariance and Generalization. Submission to CoRR 2022. https://openreview.net/forum?id=ZIlxiunnJsn
>
> [2] Zhang. Making Convolutional Networks Shift-Invariant Again. ICML 2019. https://arxiv.org/abs/1904.11486.

---

### Author Response · Authors · 2022-11-16
**General Author Response**

We thank all the reviewers for their thoughtful and highly supportive comments!

Our work makes several significant and timely contributions to the study of equivariance in neural network classifiers. Firstly, the LEE metric is a methodology contribution, with strong theoretical grounding and practical applicability. Multiple reviewers highlighted the mathematical framing in our paper as a “solid theoretical foundation”, and noted that our metric’s unique analytic layerwise decomposition was an important contribution. Beyond mathematical grounding, our metric is also proven through extensive empirical comparisons. Namely, when compared with earlier metrics, LEE often captures much more (for example aliasing effects from non-linearities) while remaining simple to implement and apply in practice.

On top of our methodological contributions, we also provide the community a large-scale empirical study on ImageNet and many popular architectures with several significant findings.
The empirical investigation was broadly acknowledged as “really quite extensive”, “solid”, “supporting their claims well”, and “providing a high quality analysis and rigorous experiments”.

Our results serve two purposes within the paper: they support the claim that LEE provides a more broadly useful tool for analysis, and deeply examines the role of a variety of design decisions in the equivariance of trained models. There are many ways to achieve model equivariance, including architectural biases, data augmentation, and effective learning of symmetries present in training data. We argue that each has a significant role to play and often it can be difficult to anticipate the exact effects of design choices without precise measurement.  We appreciate the reviewer sentiment that “this paper will inspire much future research both supporting, extending, and arguing with the hypothesis studied here.”

---

### Decision · Program_Chairs · 2023-01-20

**Decision:**

Accept: notable-top-5%

**Justification For Why Not Higher Score:**

n/a

**Justification For Why Not Lower Score:**

This is a well rounded paper, with interesting and important theoretical and empirical contributions and significant potential for impact on the direction of the field of equivariant networks.

**Metareview: Summary, Strengths And Weaknesses:**

This paper introduces a method for empirically measuring the equivariance error based on the Lie derivative, and performs an extensive empirical study. Comparing the equivariance error of different architectures such as CNNs, transformers and mixers at different scales and training methods, it is found that lower equivariance error correlates with performance, and that much of the equivariance error can be traced to spatial aliassing due to nonlinearities and other common layers. The paper also presents a very clear theoretical analysis of aliasing, equivariance error and their relation.

The reviewers unanimously agreed that this is an excellent paper, and found few significant weaknesses. I personally enjoyed reading the paper, and believe that it has significant implications for future research. So I wholeheartedly recommend the paper for acceptance.

**Note From Pc:**

if the above contains the word "oral" or "spotlight" please see: "oral" presentation means -> notable-top-5% and "spotlight" means -> notable-top-25%. As stated in our emails, we are disassociating presentation type from AC recommendations